# Clonal dynamics of alloreactive T cells in kidney allograft rejection after anti-PD-1 therapy

Garrett S. Dunlap [1,2], Daniel DiToro[2,3], Joel Henderson[4], Sujal I. Shah [2,3], Mike Manos[5], Mariano Severgnini[5], Astrid Weins[2,3], Indira Guleria[2,3], Patrick A. Ott [2,5,6,7,8], Naoka Murakami [2,9,10] ✉ & Deepak A. Rao [1,2,10] ✉

Kidney transplant recipients are at particular risk for developing tumors, many of which are now routinely treated with immune checkpoint inhibitors (ICIs); however, ICI therapy can precipitate transplant rejection. Here, we use TCR sequencing to identify and track alloreactive T cells in a patient with melanoma who experienced kidney transplant rejection following PD-1 inhibition. The treatment was associated with a sharp increase in circulating alloreactive CD8+ T cell clones, which display a unique transcriptomic signature and were also detected in the rejected kidney but not at tumor sites. Longitudinal and cross-tissue TCR analyses indicate unintended expansion of alloreactive CD8+ T cells induced by ICI therapy for cancer, coinciding with ICI-associated organ rejection.

Immune checkpoint inhibitors (ICIs) have become the standard of care therapy for many cancers[1]. ICIs block the activation of inhibitory receptors (e.g., CTLA-4, PD-1), driving T cell activation and enhancing anti-tumor immunity[2]. However, ICI therapy is often complicated by immune-related adverse events (irAEs) that result from loss of T cell tolerance[3,4]. Kidney transplant recipients have a 3- to 10-fold increased risk of cancer post-transplant, but the use of ICI is challenging due to the high risk of precipitating acute allograft rejection[5,6]. We hypothesized that ICI-induced allograft rejection might occur due to a vigorous expansion of pre-existing, alloreactive memory T cells. Here we leveraged banked tissue and blood samples from a patient with advanced melanoma who experienced allograft rejection shortly after ICI therapy to identify and track alloreactive T cells longitudinally and across different tissues.

## Results

A 77-year-old man with end-stage kidney disease due to chronic glomerulonephritis underwent deceased kidney transplantation, with six out of six HLA mismatches (A, B, and DR loci) and without any anti-HLA antibodies. His post-transplant course had been uncomplicated and without any rejection episodes, and immunosuppression was maintained with sirolimus (2 mg per day), mycophenolate mofetil (500 mg twice a day), and prednisone (5 mg per day). Baseline serum creatinine (Cr) was 1.3 mg/dl [normal range, 0.7–1.3], estimated glomerular filtration ratio (eGFR) was 50 ml per minute per 1.73 m$^2$ of the body surface area [normal range, >60], and urinalysis did not show proteinuria. Ten years after transplantation, he was diagnosed with cutaneous melanoma arising from the nasal dorsum and left cervical lymph node metastases with extracapsular extension, stage IIIC (BRAF/NRAS/c-kit wildtype).

The patient underwent a wide local surgical excision, complete left cervical lymphadenectomy, and radiation therapy to the left neck, yet subsequently developed osseous and distant lymph node metastases (Stage IV) (Supplementary Fig. 1A). Treatment with pembrolizumab was initiated (200 mg every 3 weeks) with no reduction in immunosuppressive medications. After two doses of pembrolizumab, serum Cr increased to 2.98 mg/dl. Acute transplant rejection was suspected, and an allograft kidney biopsy was performed 5 weeks after

[1]Division of Rheumatology, Inflammation, and Immunity, Brigham and Women's Hospital, Boston, MA, USA. [2]Harvard Medical School, Boston, MA, USA. [3]Department of Pathology, Brigham and Women's Hospital, Boston, MA, USA. [4]Department of Pathology, Boston Medical Center and Boston University, Boston, MA, USA. [5]Center for Immuno-Oncology, Dana-Farber Cancer Institute, Boston, MA, USA. [6]Department of Medical Oncology, Dana-Farber Cancer Institute, Boston, MA, USA. [7]Broad Institute of MIT and Harvard, Cambridge, MA, USA. [8]Department of Medicine, Brigham and Women's Hospital, Boston, MA, USA. [9]Division of Renal Medicine, Brigham and Women's Hospital, Boston, MA, USA. [10]These authors contributed equally: Naoka Murakami, Deepak A. Rao. ✉e-mail: nmurakami1@bwh.harvard.edu; darao@bwh.harvard.edu

initiation of pembrolizumab (Fig. 1A). Pembrolizumab was discontinued, the patient received solumedrol (500 mg daily for 3 days) followed by a prednisone taper (1 mg per kilogram per day), and the dose of mycophenolate mofetil was increased (1000 mg twice a day). Serum Cr level initially improved to 2.2 mg/dl but 2 weeks later increased again to 3.1 mg/dl. The patient received another course of solumedrol (500 mg daily for 3 days), followed by prednisone taper, resulting in Cr stabilization between 1.7 and 1.9 mg/dl (Fig. 1A). No irAEs were observed. Restaging PET/CT 12 weeks after initiation of pembrolizumab demonstrated a mixed response of lymph node and bone metastases (Supplementary Fig. 1B, C).

Histopathological analysis of the allograft biopsy showed prominent interstitial inflammation (Banff score, i3), moderate glomerulitis (g2), mild tubulitis (t1), and no identified vasculitis (v0) (Fig. 1B, C)[7]. C4d staining was negative, and the donor-specific antibody titer was 0, indicating an absence of antibody-mediated rejection. Immunofluorescence staining of the allograft biopsy indicated infiltration of both CD4[+] and CD8[+] T cells, suggesting acute T cell-mediated rejection (Fig. 1D).

Given that the recipient's graft post-transplant course was stable until ICI therapy, we hypothesized that ICI therapy had activated a population of pre-existing alloreactive T cells. To identify these alloreactive T cells in vitro, we performed a mixed-lymphocyte reaction (MLR) of recipient PBMCs and donor splenocytes, which had been previously banked. Recipient T cells proliferating after MLR were flow-sorted, and droplet-based single-cell RNA sequencing (scRNA-seq) was used to obtain paired gene expression and T cell receptor (TCR) sequences for the sorted cells (Supplementary Fig. 2A). After dimensionality reduction of the 1505 sorted cells that passed quality control

filtering, we identified nine clusters, including populations of CD8[+] and CD4[+] T cells (Fig. 2A, Supplementary Fig. 2B–F, Supplementary Data 1). Reasoning that the cells with the greatest alloreactive potential would show the highest proliferative capacity, we determined the proliferative signature of each cluster using a previously reported gene set (Supplementary Data 2)[8]. Two clusters of CD4[+] T cells (C3 and C5) and one cluster of CD8[+] T cells (C1) had elevated proliferation signatures compared to all other populations (Supplementary Fig. 3A, B). Further, analysis of the expression patterns of cell cycle phase-associated genes suggested that cells contained in these three clusters were actively cycling, whereas the majority of cells from all other clusters were predicted to be noncycling and in the G1 phase (Fig. 2B, Supplementary Fig. 3C, D, Supplementary Data 2). Among noncycling clusters, C2 showed features of cytotoxicity (GZMB, PRF1, GNLY, GZMH), C7 had regulatory T cell markers (FOXP3, IKZF2), and C8 showed a high interferon signature (MX1, ISG16, OAS1) (Supplementary Fig. 2C, D).

We then interrogated the paired TCR information available for 1311 (87.1%) of these T cells. Both CD8[+] T cell clusters (C1 and C2) contained expanded clones, while few expanded clones were found among CD4[+] T cell populations (Fig. 2C and Supplementary Fig. 4A). Notably, nine of the 12 largest T cell clones identified by MLR localized to cluster C1 (Fig. 2D). We did not observe clonal sharing between C1 and C2, indicating that these expanded T cell populations are not clonally related (Supplementary Fig. 4B). Given the MLR clone localization and elevated proliferation signature of C1, we posit that this cluster represents bona fide alloreactive CD8[+] T cells.

To track the alloreactive T cell clones over time and in different tissues, we performed bulk TCR sequencing on patient-derived PBMCs obtained before, during, and after pembrolizumab treatment, as well

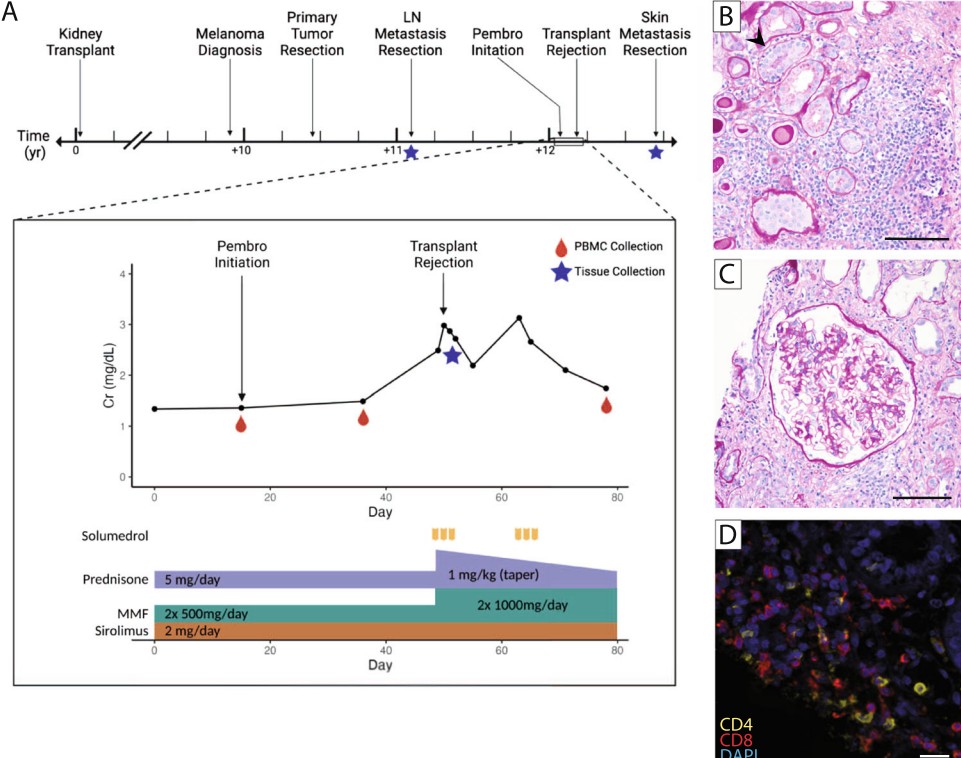

**Fig. 1 | Case of kidney transplant rejection following immune checkpoint inhibitor treatment. A** Timeline of events, including notation of samples collected. The callout box denotes the timeline of the medication regimen and creatinine (Cr) levels during immune checkpoint inhibitor therapy. **B**–**D.** Histology of immune cell infiltration in the allograft biopsy. **B** Periodic acid–Schiff (PAS) stain of allograft biopsy showing prominent interstitial inflammation (i3), focal tubulitis (arrowhead, t1), and proteinaceous cast in tubules (200x). **C** PAS stain of allograft biopsy showing glomerulus with moderate glomerulitis (g2) and mesangial expansion (200x). **D** Immunofluorescence staining of CD4 (yellow), CD8 (red), and DAPI (blue). Scale bars for (**B**) and (**C**), 100 μm. Scale bar for **D**, 20 μm. A single replicate was performed for each staining in (**B**), (**C**), and (**D**). MMF mycophenolate mofetil.

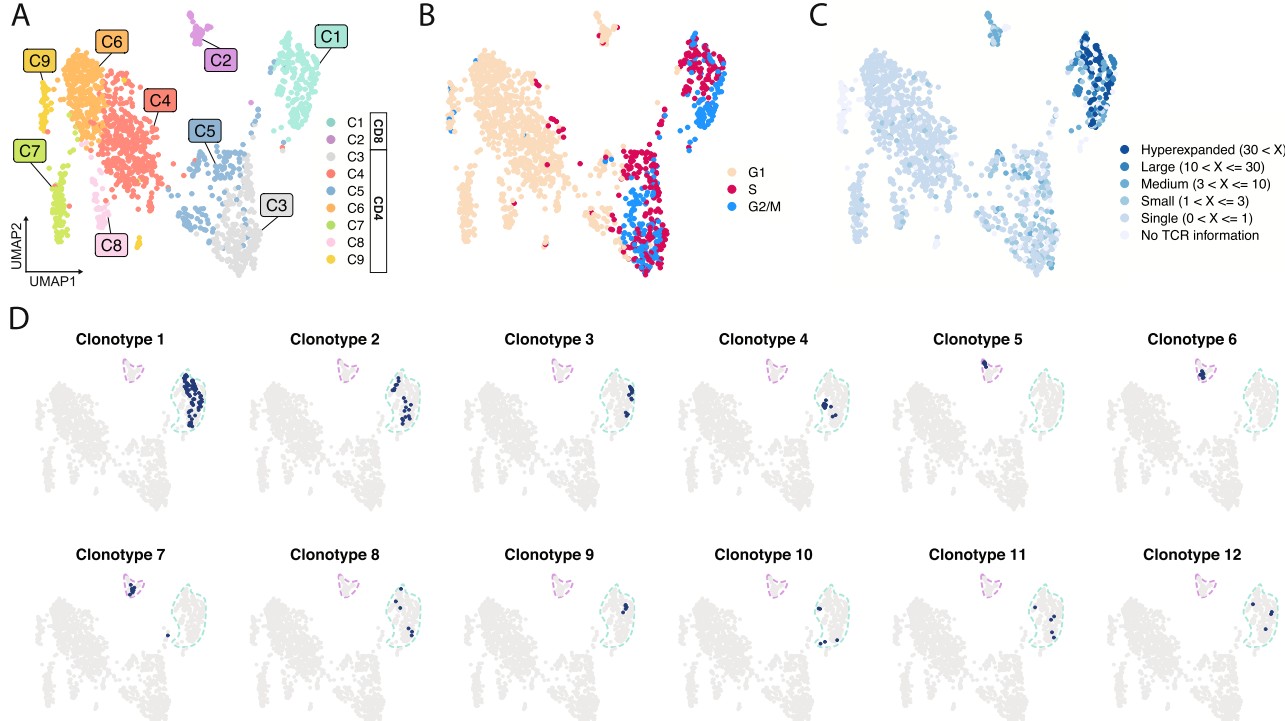

**Fig. 2 | Mixed-lymphocyte reaction identifies highly proliferative and clonally expanded alloreactive CD8⁺ T cells. A** Uniform Manifold Approximation and Projection (UMAP) clustering of T cells isolated from MLR. **B** UMAP of predicted cell cycle phase of T cells from MLR. **C** UMAP of T cells from MLR showing expanded clones by category. **D** UMAP localization of the top 12 individual clones. Clusters C1 (green) and C2 (purple) are outlined.

as on kidney allograft biopsy, pre-pembrolizumab metastatic tumor-containing lymph node, and post-pembrolizumab metastatic tumor-containing skin samples (Fig. 1A, Supplementary Fig. 5). A decrease in overall clonal T cell diversity in blood after initiation of treatment with pembrolizumab (Pembro 1) suggests T cell activation in response to ICI therapy, as has been previously described in relation to clinical response and irAE severity (Fig. 3A, Supplementary Fig. 5)[9,10]. Flow cytometric analysis of the PBMC samples showed a sharp increase in Ki67⁺ CD8⁺ T cells, confirming the activation of T cells (Supplementary Fig. 6A, B). We also observed an increased frequency of the T effector memory RA (TEMRA) population and a decreased frequency of the naive population among CD8⁺ T cells at this timepoint (Supplementary Fig. 6C). Assessing for clonal overlap between samples, we found much higher clonal similarity among T cells in serial PBMCs compared to those in kidney, skin, and lymph node tissue. However, among these tissues, T cells in the kidney exhibited the highest degree of clonal overlap with those in PBMCs (Fig. 3B).

We next examined whether alloreactive clones identified by MLR could be found in the tissue samples. Strikingly, the alloreactive T cell clones were present in the rejected kidney but not in the tumor-containing lymph node or skin samples, indicating preferential trafficking of alloreactive clones into the allograft (Fig. 3C, D, Supplementary Fig. 7). Further, alloreactive T cell clones were absent in pre-pembrolizumab PBMCs but increased dramatically in post-pembrolizumab PBMCs, consistent with the expansion of circulating alloreactive clones following pembrolizumab treatment.

We further found a substantial difference in clonal frequencies of MLR clones in the alloreactive T cell cluster (C1) and in cluster C2 (Fig. 3E). While alloreactive clones could be found at relatively low frequencies in the blood early after pembrolizumab initiation and varied in abundance between tissue compartments, clones from C2 were found at high numbers in nearly all samples and timepoints. When comparing the frequencies of clones between the MLR and bulk TCR samples, these C2 clones were found to be present in relatively

similar proportions in the MLR and PBMCs, suggesting that these cells did not expand or expand only minimally. This pattern could be consistent with the presence of a set of high-frequency viral-reactive TCRs across tissues, including in the kidney[11]. In support of this hypothesis, examination of a database of public TCRs revealed a clone associated with reactivity to an influenza peptide within C2 (Supplementary Data. 3). Further, a comparison of C1 and C2 cells using multiple recently-identified signatures of viral-reactive T cells showed elevated scores across C2 clones (Supplementary Fig. 8)[12,13]. In contrast to the C2 clones, we observed a marked increase in the frequency of alloreactive C1 clones between the MLR and bulk TCR samples, further highlighting the efficacy of our MLR approach to expand alloreactive clones that may have mediated kidney transplant rejection in this patient (Fig. 3F).

After identifying and tracking alloreactive clones across tissues and over time, we next aimed to define the phenotypes of these alloreactive cells in the blood after pembrolizumab therapy. We sorted non-naive CD8⁺ T cells from both post-pembrolizumab blood samples and performed scRNA-seq to generate paired RNA and TCR libraries. Following clustering of the 14,186 CD8⁺ T cells from both samples, a combination of differential-gene expression analysis and comparison to recently published blood CD8⁺ T cell reference datasets was used to identify subpopulations (Fig. 4A, B, Supplementary Fig. 9A, B, Supplementary Data 4)[14–16]. Expected populations of cytotoxic T lymphocytes (CTLs), effector memory (EM), central memory (CM), and mucosal-associated invariant T (MAIT) cells were present. We also identified a population of cells marked by the strong expression of *ZNF683* (encoding Hobit), *CXCR3*, and *HLA-DRA*, which we termed ZNF683⁺ T cells (Fig. 4B). This cluster showed an elevated activation signature, potentially suggesting a set of cells that have become activated with the onset of ICI therapy (Fig. 4C, Supplementary Data 2). Clustering analysis further identified small populations of actively proliferating cells and cells with elevated mitochondrial gene expression (Fig. 4B, Supplementary Fig. 9B). Deeper analysis of the proliferating cluster indicated a majority of included cells belong to the

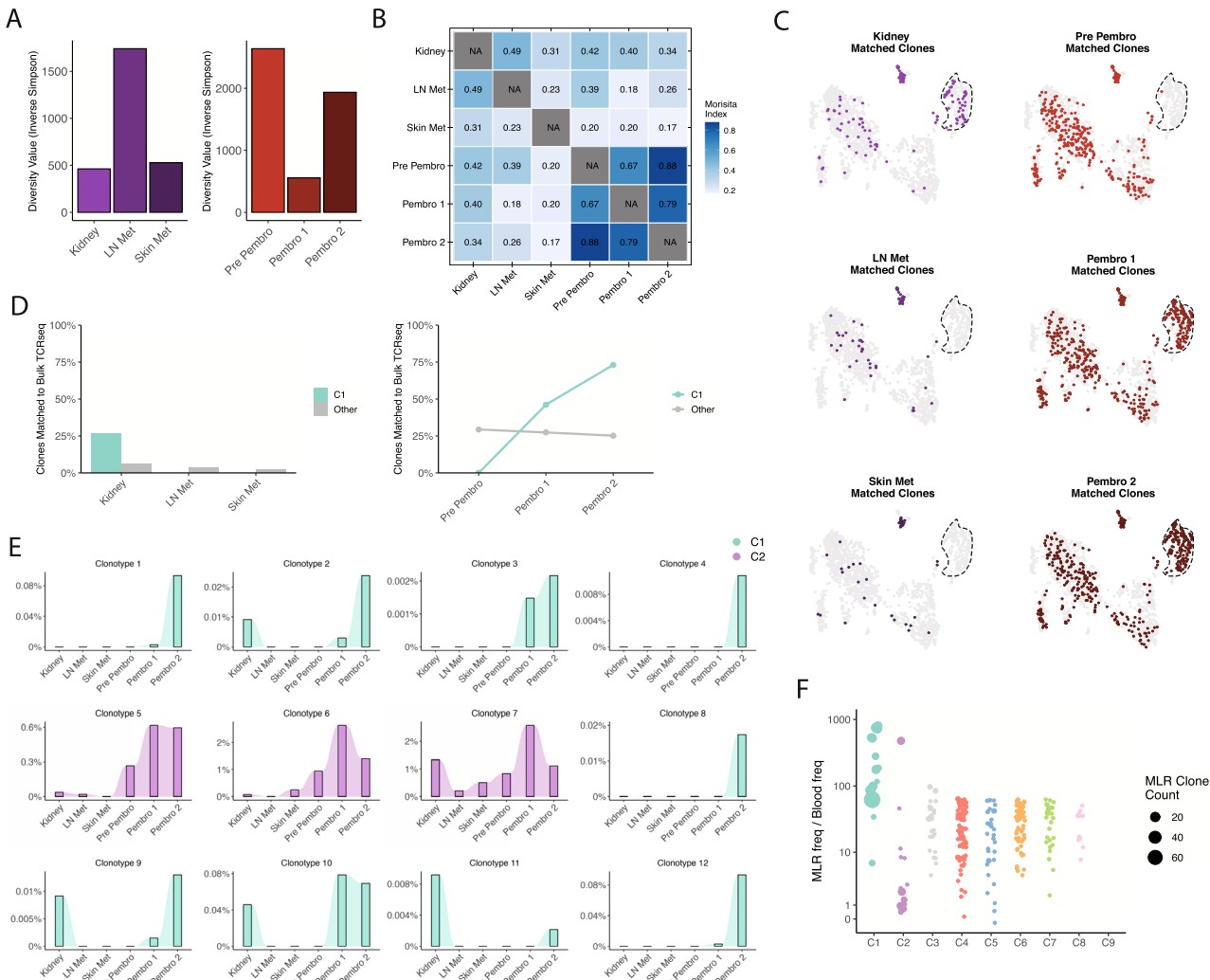

**Fig. 3 | Tracking alloreactive clones across tissue and blood. A** Bar plot of Inverse Simpson diversity index for tissue (left) and blood (right) bulk TCR sequencing samples. **B** Heat map of clonal sharing by Morisita overlap index between bulk TCR sequencing samples. **C** UMAP visualization of MLR T cells colored by whether they were found in indicated bulk TCR dataset. A dashed border is drawn around cluster C1. **D** Percentage of C1 clones that match a clone in indicated bulk TCR dataset of tissue (left) and blood (right). **E** Among bulk TCRs from indicated sources, percentage of TCRs represented by indicated clone. The color indicates the originating cluster (C1 is green, C2 is purple) for each clonotype. **F** Fold expansion between MLR and PBMC TCR sequencing samples for each MLR clonotype, split by MLR cluster. The dot size indicates MLR clone size.

ZNF683[+] subpopulation (Supplementary Fig. 9D). Analysis of the composition of these populations between timepoints was consistent with our previous flow cytometry analysis, with increased CTLs and proliferating cells, and decreased EM cells, at Pembro 1 compared to Pembro 2 (Fig. 4D, Supplementary Figs. 6B, C, 9E).

Leveraging the paired TCR information for 13,034 (91.8%) of these non-naive CD8[+] T cells, we sought to detect the cluster localizations of the MLR-defined alloreactive clones. Clones matching the alloreactive C1 MLR cluster (13/19) could be identified across both timepoints. Strikingly, at the Pembro 1 timepoint, these clones mapped predominately to the proliferating and ZNF683[+] clusters, while at the Pembro 2 timepoint, the matching clones belonged almost exclusively to the ZNF683[+] cluster (Fig. 4E, F). In contrast, clones matching the C2 MLR cluster (12/15) were associated predominantly with CTLs but could be identified in all clusters, with proportions that did not change across timepoints (Fig. 4G, H). These results indicate that the MLR-defined alloreactive T cell clones possess a specific transcriptomic in the circulation that distinguishes them from the majority of circulating T cells.

To extend this analysis beyond the MLR-detected clones, we evaluated the phenotype of T cell clones that had newly emerged in the blood after pembrolizumab treatment using the bulk TCR data. Remarkably, clones were newly detected in the post-pembrolizumab blood samples but not present in pre-pembrolizumab samples (Emerged) and were highly overrepresented in the ZNF683[+] cluster compared to clones that were stably present before and after pembrolizumab treatment (Fig. 4I, J). Together, these highlight a specific phenotype of identified alloreactive clones as they become activated and enter the circulation following ICI initiation.

## Discussion

By combining single-cell profiling of in vitro MLR-expanded alloreactive CD8[+] T cells, bulk TCR sequencing of longitudinally collected tissue specimens, and paired scRNA-seq/TCR-seq of blood CD8[+] T cells, we demonstrate here that alloreactive CD8[+] T cells clonally expanded after ICI therapy, acquired a distinctive transcriptomic signature, and accumulated in the transplanted kidney during anti-PD-1-mediated allograft rejection. These data provide key insights into the unintended expansion of pathologic T cells induced by ICI cancer therapy.

Our observations suggest that despite 10 years of stable allograft function, alloreactive T cells remained present in the kidney transplant recipient, perhaps due to a combination of immunosuppression and

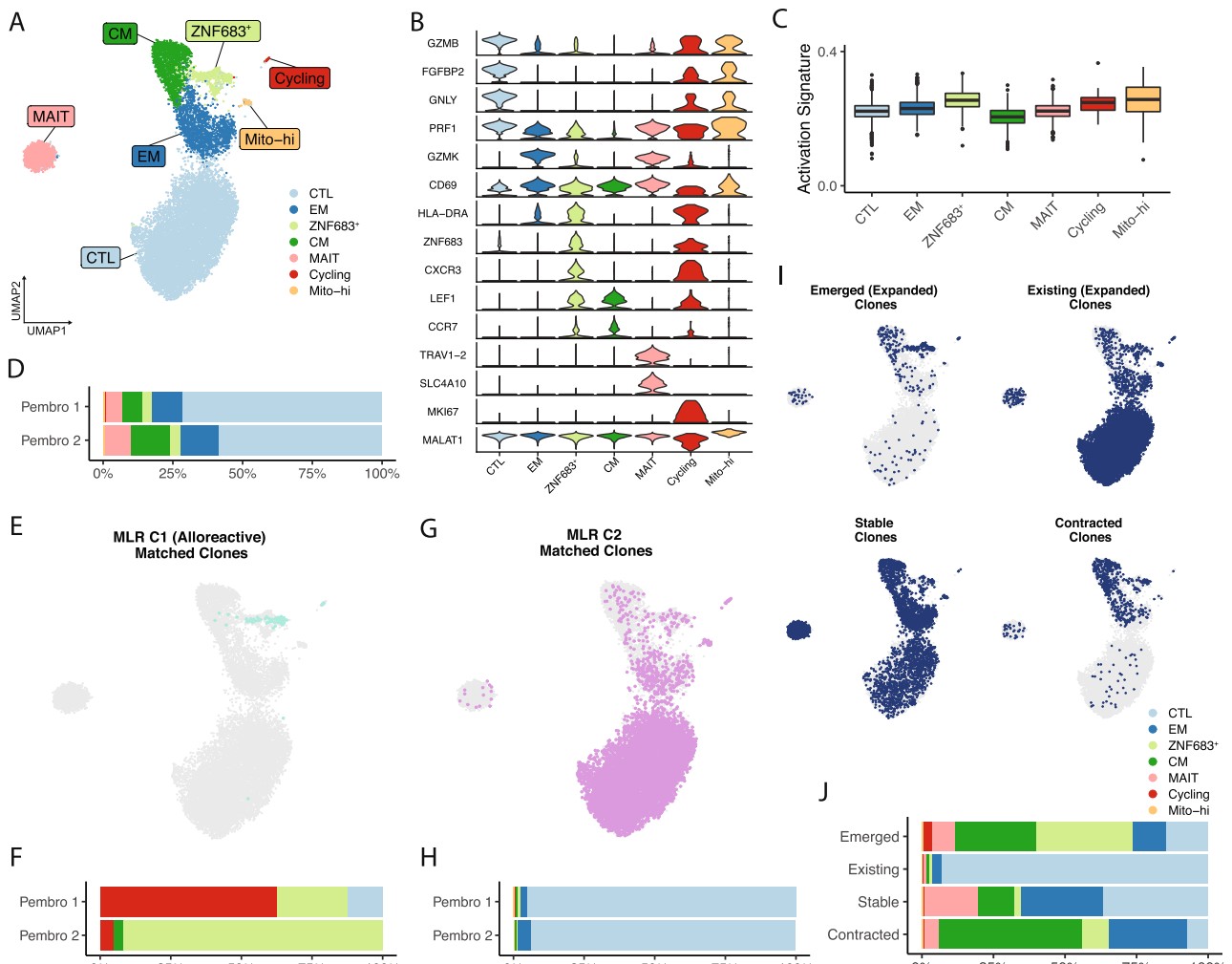

**Fig. 4 | Deciphering the phenotypes of alloreactive clones in the blood. A** UMAP clustering of non-naive CD8+ T cells isolated from post-pembrolizumab blood samples. **B** Violin plot of cluster-defining markers. **C** Box plot of activation signature across clusters. **D** Bar plot of cluster proportions across timepoints. **E** UMAP of MLR C1 matching clones (green dots). **F** Bar plot of the cluster distribution of MLR C1 matching clones across timepoints. **G** UMAP of MLR C2 matching clones (purple dots). **H** Bar plot of the cluster distribution of MLR C2 matching clones across timepoints. **I** UMAPs highlighting clones expanded or contracted in blood. **J** Bar plot of the cluster distribution of clones expanded or contracted in the blood. Boxes in (**C**) denote the interquartile range, with the horizontal line at the median, and outlier cells are shown as dots.

peripheral tolerance mechanisms. The data are consistent with prior TCR sequencing analyses, which demonstrated clonal deletion of donor-reactive T cells in tolerant kidney transplant patients after combined bone marrow–kidney transplant, but not in conventional kidney transplant patients on immunosuppression[17]. In the patient reported here, ICI therapy, even in the presence of an uninterrupted immunosuppression regimen, appears to have unleashed the alloreactive T cell pool from PD-1-mediated regulation, resulting in an accumulation of alloreactive T cells in the circulation and in rejection of the kidney allograft.

It is difficult to define the phenotype of the alloreactive T cells expanded in an MLR because features will change with in vitro culture and activation; therefore, we used parallel scRNA-seq/TCR-seq of CD8+ T cells from the same blood samples to define the ex vivo phenotypes of T cells from the same clones. The MLR-defined alloreactive clones displayed a remarkably homogeneous phenotype, characterized by expression of *CXCR3* and *ZNF683*, with a transcriptomic signature that clearly distinguished these cells from the majority of circulating CD8+ T cells, including circulating cytotoxic, effector memory, and central memory cells. Beyond the MLR-defined clones, a larger set of clones that were detected in blood only after pembrolizumab also preferentially mapped to the ZNF683+ cluster, suggesting that this

transcriptomic signature may capture a circulating set of cells that is characteristic of ICI therapy. The expression of *ZNF683* in this cluster raises the possibility that these cells may be derived from tissue-resident cells or possess some features of tissue residency[18–20]. Future studies may help define the phenotypes of these alloreactive T cells prior to ICI therapy and determine whether they are resident within the allograft before ICI-induced activation or are mobilized from other tissue compartments.

The MLR approach used here identified a subset of alloreactive T cells, likely prioritizing highly proliferative CD8+ T cells, yet modifications to the stimulation strategy may allow the detection of additional alloreactive T cell populations. As the clonal frequency of alloreactive T cells is below detection at the pre-ICI timepoint in our experiments, it remains unclear whether an MLR preceding ICI therapy can expand and identify alloreactive T cells to predict future acute rejection upon exposure to ICI. Further, the advancement of techniques to identify antigen targets of T cells may enable empiric discovery of T cell epitopes and pinpoint antigens that drive such rejection responses[21,22].

While our work is a retrospective study in one patient, the kinetics of rejection in transplant patients treated with ICI therapy lends itself to similar clonal tracking across time in a prospective cohort of

patients. Rapid acute rejection following ICI therapy has been reported throughout solid organ transplants, such as kidney, liver, heart, and lung[23], and future studies inclusive of multiple sites of organ transplant may further shed light on if such dynamics can be observed in other organ systems. Several prospective clinical studies are investigating strategies to mitigate acute rejection in kidney transplant recipients treated with ICI: continuing tacrolimus (NCT03816332), conversion to mammalian target of rapamycin (NCT04339062), or maintaining the baseline immunosuppression regimen[24,25]. It is anticipated that results from these trials will help illuminate the underlying mechanisms of ICI-associated acute allograft rejection.

Overall, this work establishes a framework for the detection and tracking of pathogenic T cell clones expanded by ICI therapy using TCR sequencing. These methods may enable a deeper understanding of the mechanisms of tumor immunity and irAEs after ICI therapy, with the ultimate goal of uncoupling these two.

## Methods

### Patient samples
This study was approved by the listed research ethics committees, and the patient consented to participate in the collection of biomedical specimens. The biobank study has been approved by Dana Farber Cancer Institute Institutional Review Board (DFCI IRB 05-042). Heparinized blood samples were obtained at the time of ICI initiation and longitudinally thereafter. Peripheral blood mononuclear cells (PBMC) were isolated using Ficoll/Hypaque density-gradient centrifugation (GE Healthcare) and cryopreserved in 10% DMSO-FCS (Sigma-Aldrich) in liquid nitrogen. Skin, lymph node, and allograft kidney biopsies were obtained for clinical indication and subjected to formalin fixation and paraffin embedding (FFPE). Donor splenocytes were cryopreserved in 10% DMSO-FCS and archived at BWH Tissue Typing Laboratory (BWH IRB 2021P003483).

### Flow cytometry
PBMCs were thawed and stained with fixable viability dye (Thermo Fisher) for 30 min on ice, followed by surface staining in 2% FCS-PBS with various antibodies for 30 min on ice. For intracellular cytokine staining, the cells were stimulated with phorbol myristate acetate (PMA) and ionomycin in the presence of GolgiStop (BD, #554724) for 4 h at 37 °C, 5% $CO_2$, and stained with cell surface markers, followed by permeabilization with Foxp3 staining kit (eBioscience, #00-5523-00) and intracellular staining. The samples were analyzed by flow cytometry (FACS Canto-II, BD) immediately after staining using FACSDiva software (BD, v9.0). The data were analyzed by FlowJo software (BD, v10), and data were graphed using GraphPad Prism (v9). The list of antibodies used is provided in Supplementary Data 5.

### Immunofluorescence
Formalin fixed, paraffin embedded (FFPE) tissue samples were sectioned, and slides were deparaffinized, blocked (Bloxall, Vector Laboratories, SP-6000-NB), incubated with anti-CD4 (clone GR3276764-2, Abcam, EPR6855) or anti-CD8 (clone GR208681-1, Abcam, ab85792) followed by a secondary antibody (Opal Polymer HRP anti-mouse + anti-rabbit, Akoya, ARH1001EA) then dye (TSA Cy3, Akoya, NEL744001KT or TSA Cy5, Akoya, NEL745001KT). The samples were mounted with ProLong Diamond Antifade mountant containing DAPI (Invitrogen, P36970). Images of the tissue specimens were acquired using the TissueFAXS platform (TissueGnostics) at 71X magnification.

### Bulk TCR sequencing
Samples were sent to Adaptive Biotechnologies for bulk TCRb sequencing using the immunoSEQ assay. Acquired data for each sample were exported from the immunoSEQ Analyzer platform, including CDR3 sequences, sequence read counts for each clone, and information on identified V, D, and J genes. Exported data was then loaded into R (v4.0.2) for further analysis.

### Mixed-lymphocyte reaction
Donor splenocytes were gamma-irradiated (30 Gy) and were loaded with CellTrace Violet dye (Invitrogen, C34557) to be able to distinguish donor and recipient cells. Then, $1 \times 10^6$ recipient PBMC (Post-Pembro 2 timepoint) were loaded with CFSE (Invitrogen) and cocultured with $4 \times 10^6$ irradiated donor splenocytes for 5 days in 10% FCS-RPMI, supplemented with recombinant human IL-2 (20 units/ml, PeproTech) and anti-CD28 (1 μg/ml) at 37 °C and 5% $CO_2$. After co-culture, the cells were stained with fixable viability dye, CD3, CD4, and CD8. The viable/violet$^-$/CD3$^+$/CFSE$^{low}$ populations were flow-sorted using FACS Aria (BD) and used for single-cell RNA and TCR sequencing.

### Single-cell RNA and TCR library preparation and sequencing
Sorted cells were immediately encapsulated in oil droplets using the 10x Genomics Chromium instrument at the Brigham and Women's Hospital Center for Cellular Profiling. Single-cell 5′ RNA transcriptome and V(D)J libraries were then constructed using a Chromium Single Cell Immune Profiling Reagent Kit (v2, #1000263) following the manufacturer's instructions. Pooled RNA and TCR libraries were sequenced on an Illumina NovaSeq 6000.

### MLR single-cell RNA processing, QC, and clustering
Sequencing reads for the RNA library were processed through the CellRanger workflow (v6.0.1). Briefly, BCL files from the sequencing output were used to generate FASTQ files using the CellRanger mkfastq command. Reads were then aligned to a reference (GRCh38), filtered, and counts of barcodes and UMIs were generated using the CellRanger count command. The filtered_feature_bc_matrix.h5 file output from the pipeline was loaded into Seurat (v4.0.2) for downstream analysis. Seurat was used for quality control filtering, where cells with less than 25% of reads associated with the mitochondrial genome and cells with greater than 200 features were retained. In the remaining cells, the data were log-normalized and scaled before dimensionality reduction using Uniform Manifold Approximation and Projection (UMAP) with the first 30 principal components (PCs) was applied. After a systematic assessment of the output of varying cluster resolution, a value of 0.7 was selected for final cluster generation. Initial differential gene expression between clusters was determined using a Wilcoxon rank-sum test, with the resulting list filtered to only retain genes with a log fold-change greater than 0.25 compared to all other clusters and those which were detected in at least 20% of the population being tested.

### Non-naive CD8 single-cell RNA processing, QC, and clustering
Initial processing of the sequencing reads using CellRanger was completed as above. Quality control filtering was applied to retain cells with less than 15% of reads associated with the mitochondrial genome and cells with between 200 and 5000 features. Log-normalization and scaling were then performed, with VDJ genes removed as potential variable features. Harmony[26] was used to remove batch effects associated with timepoint and clone (thetas = 2 and 0.1, respectively). Using the top 20 embeddings from Harmony, clustering and dimensionality reduction using UMAP were performed. Similar to the above, varying cluster resolutions were tested, and a final resolution of 0.2 was employed.

### Gene signature analysis
Gene signatures for proliferation, activation, cell cycle phase, and viral specificity were obtained through the references listed in Supplementary Data 2. Scores for each cell were obtained using the

AddModuleScore function in Seurat, which calculates the average expression of all genes contained in the signature.

### Reference mapping

Reference objects were built using Symphony[27] with blood CD8+ T cells from multiple sources[14–16], which were independently integrated at the sample level using the first 20 PCs for each. Non-naive CD8+ T cells from this study were subsequently projected onto these references using the mapQuery and knnPredict functions. Following the generation of confidence scores for each reference cluster's mapping, we visualized all mappings together using pheatmap.

### Single-cell TCR processing and QC

Sequencing reads for the TCR library were processed through the CellRanger workflow (v6.0.1), with BCL files used to generate FASTQ files using CellRanger mkfastq, as above. Reads were then aligned to a TCR reference (vdj-GRCh38), productive contigs were filtered, and CDR3 regions were identified for each cell. Cells with the same V(D)J transcripts were then grouped into the same clonotype. The filtered_contig_annotations.csv output from CellRanger was loaded into R, and cells with >2 chains were filtered to retain the alpha and beta chains with the highest expression using scRepertoire (v1.1.4). The TCR for each cell was then paired with its corresponding transcriptomic data through matching cell barcodes.

### Statistics and reproducibility

As this study focuses on the case of a single patient, no statistical method was used to predetermine the sample size. Further, no data were excluded from the analyses, the experiments were not randomized, and the investigators were not blinded to allocation during experiments and outcome assessment.

### Reporting summary

Further information on research design is available in the Nature Portfolio Reporting Summary linked to this article.

## Data availability

The processed single-cell RNA/TCR and bulk TCR sequencing data used in this study are available in the NCBI Gene Expression Omnibus (GEO) database under accession code GSE216763. The bulk TCR sequencing data used in this study are also available in the Adaptive Biotechnologies ImmuneACCESS portal [https://doi.org/10.21417/2022NC]. Source data are provided with this paper.

## Code availability

Scripts used for the analyses and figure generation of this paper are available at https://github.com/dunlapg/transplant-rejection-ICI.

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

## Acknowledgements

We thank the BWH Center for Cellular Profiling for assistance with cell sorting and single-cell RNA/TCR library preparation. This work was supported in part by funding from the National Institutes of Health (NIH): K08-AR072791 (D.A.R.), P30-AR070253 (D.A.R.), K08-DK120868 (N.M.), R03-DK131223 (N.M.), R01-CA229261 (P.A.O.); as well as a Burroughs Wellcome Fund Career Award in Medical Sciences (D.A.R.) and American Society of Nephrology Carl W. Gottschalk Research Scholar Award (N.M.).

## Author contributions

Conceived and designed the analysis: G.S.D., D.D., N.M., and D.A.R. Collected and processed clinical samples: J.H., S.I.S., M.M., M.S., A.W., I.G., and P.A.O. Completed MLR and flow cytometry: D.D. and N.M. Performed computational analysis: G.S.D. Writing—original draft: G.S.D. and N.M. Writing—review and editing: G.S.D., D.D., P.A.O., N.M., and D.A.R. Supervised the work: N.M. and D.A.R.

## Competing interests

P.A.O. has received research funding from and has advised Neon Therapeutics, Bristol-Myers Squibb, Merck, CytomX, Pfizer, Novartis, Celldex, Amgen, Array, AstraZeneca/ MedImmune, Armo BioSciences, Evaxion, Oncorus, Xencor, and Roche/Genentech.
