## [Peer Review File · Nature Communications]

Clonal dynamics of alloreactive T cells in kidney allograft rejection after anti-PD-1 therapyREVIEWER COMMENTS

Reviewer #1 (Remarks to the Author):

Dunlap and colleagues elegantly describe their observations about the clonal dynamics of alloreactive T cells in a kidney transplant recipient who received anti-PD-1 for treatment of advanced melanoma. These data are of great interest in this emerging area of immuno-oncology.

The authors describe a 77-year-old man who underwent kidney transplantation 10 years prior to developing unresectable BRAF/cKIT/NRAS-WT melanoma. It would be helpful to better characterize the tumor response to pembro using validated criteria (e.g., RECIST v1.1). The inclusion of representative CT scan images may be helpful.

The authors point out that the ultimate goal is to uncouple anti-tumor and anti-self/allograft responses. The manuscript might benefit from a brief discussion of how ongoing prospective trials (e.g., Schenk KM, et al. Nivolumab (NIVO) + tacrolimus (TACRO) + prednisone (PRED) +/- ipilimumab (IPI) for kidney transplant recipients (KTR) with advanced cutaneous cancers. *Journal of Clinical Oncology* 2022 40:16_suppl, 9507-9507) might incorporate the strategies employed in the current study in pursuit of that objective.

Reviewer #2 (Remarks to the Author):

In this manuscript, Deepak A. Rao and co-workers have identified and tracked alloreactive T cells in a patient with melanoma who suffered from kidney transplant rejection following ICI therapy with the TCR sequencing. The results indicated that ICI therapy was related with an increase in circulating alloreactive CD8+ T cell clones. However, only TCR sequencing was performed. The conclusion was not reliable. More details should be added to investigate the mechanism about the increase of the CD8+ T cell clones.

Reviewer #3 (Remarks to the Author):

This manuscript describes the expansion and distribution of alloreactive t-cells in a formally stable kidney transplant patient who develops cellular rejection after treatment with the ICI Pembrolizumab for malignant melanoma. T-cell clusters with an increased proliferation signal were identified using single cell sequencing of MLR expanded recipient PBMC's. An allo specific CD8 T cell clone was identified within cluster 1. Via TCR sequencing alloreactive clones were found to expand in peripheral blood and in the rejected kidney but not in a melanoma positive skin or lymph node biopsy.

This paper provides an interesting approach to the investigation of alloreactive t-cell dynamics in rejection in general and in specific for rejection induced by ICI. The potential of MLR to identify allo specific clones that can be followed in periferal blood is shown. This method offers a number of opportunities to further our understanding of the dynamics of alloreactivity. While the study is limited by the fact that only a single case is described and the findings are within the range of what would have been expected, the paper does show and interesting application of single cell technology in understanding rejection and may lead to meaningful follow up.

I do have a few questions.

1. I could not find a statement on patient consent and ethics committee approval. I may have missed this. Otherwise please include this.
2. The timing of the recipient sample for the initial MLR experiment is not clear to me. When were the samples obtained?

3. The authors do not mention the possible clonal expansion of t-cells directed against the tumour. Did the authors also identify the expansion of clones that were not explained by alloreactivity but possibly by increased anti-tumor immunity?

4. Do the authors envision that a initial MLR would help to predict the risk of rejection in patients who receive ICI?

Reviewer #4 (Remarks to the Author):

This study involves investigating alloreactive memory T cells in a patient with advanced melanoma who experienced allograft rejection after immune checkpoint inhibitor (ICI) therapy. The authors used single cell RNA and TCR sequencing, plus bulk RNA sequencing of TCR, to identify and track alloreactive T cells clones throughout the patients PD-1 treatment for melanoma. The authors conclude that they were able to identify a post ICI therapy subset of clonally expanded alloreactive CD8+ T cells.

What is noteworthy and unique about this particular study is that the tissues comes from a human patient with a clinically interesting case: stable conventional kidney transplant that develops advanced melanoma, is treated with ICI therapy and results in rejection of the transplant. Tissue samples were taken at key time points throughout the patients PD-1 therapy to longitudinally measure T cell clonality.

In Figure 2A the authors show a UMAP of single T cells collected from the MLR reaction – it shows 9 clusters that are grouped by CD8+ and CD4+ markers based on data they show in Supplemental data 1B-D. They then do further analysis using cell cycle markers to identify the proliferating CD8+ (cluster 1) population which they predict to be the alloreactive T cells. It is mentioned in the methods that differential expression with (Seurat-based FindMarkers function) and some of these markers are shown in Supplemental Figure 1B. I think the authors should show more of the DE genes (at least the top 5-10) as a heatmap and assign or discuss the biological significance of the remaining clusters. Especially Cluster 2 – it appears to be quite different from the other T cell populations. The cluster 2 population is referenced throughout the rest of the study (Figure 3, Supplemental Figure 7) so it would be helpful to try to describe the biological significance.

The colors of the clusters are difficult to distinguish in the UMAPs in Figure 2A and Supplemental Figure 2B. Otherwise, I would label the clusters directly. For the UMAPs in Figure D2 showing expanded clonotypes, it would be helpful if cluster 1 and cluster 2 (major CD8+ populations) be outlined – similar to Figure 3C.

Basic statistics for single cell RNA and TCR sequencing should be included in the methods or supplemental data: How many total cells are in this one sample? What is the percent of cells assigned clonotypes to single cells? What percentage of cells have an assigned clonotype for each cluster (resolution 0.7)?

The code for the single cell RNA analysis and TCR analysis, and the bulk TCR analysis should be submitted as supplementary information.

Database accession number for single cell RNA and TCR sequencing and bulk TCR sequencing was not provided with the manuscript.

REVIEWER COMMENTS

Reviewer #1 (Remarks to the Author):

Dunlap and colleagues elegantly describe their observations about the clonal dynamics of alloreactive T cells in a kidney transplant recipient who received anti-PD-1 for treatment of advanced melanoma. These data are of great interest in this emerging area of immunology.

Response: We thank the reviewer for their comment and interest.

The authors describe a 77-year-old man who underwent kidney transplantation 10 years prior to developing unresectable BRAF/cKIT/NRAS-WT melanoma. It would be helpful to better characterize the tumor response to pembro using validated criteria (e.g., RECIST v1.1). The inclusion of representative CT scan images may be helpful.

Response: The reviewer raises an important point. We note that the patient was not treated on a clinical trial and therefore tumor response was not assessed prospectively using RECISTv1.1. Instead, the patients underwent serial PET/CTs. Overview images of the PET portion of these scans are shown in Supplemental Figure 1 (included below). The initial tumor assessment by PET/CT revealed a mixed response - outlined below.

“Representative PET scan images obtained 2 weeks prior to treatment with pembrolizumab (A), 3 months after pembrolizumab initiation (B), and 6 months after pembrolizumab initiation (C) are shown. Between (A) and (B), there was interval mixed response with new FDG-avid cervical lymph nodes and persistent or increasing FDG uptake in some bone lesions and periportal lymph nodes and decrease in extent and intensity of other bone lesions. There was stable intense cutaneous uptake in the left neck. Between (B) and (C), there was a mixed response with the appearance of a new FDG-avid left axillary node and decreased FDG uptake in multiple lymph nodes. There was persistent cutaneous uptake in the left neck.”

The authors point out that the ultimate goal is to uncouple anti-tumor and anti-self/allograft responses. The manuscript might benefit from a brief discussion of how ongoing prospective trials (e.g., Schenk KM, et al. Nivolumab (NIVO) + tacrolimus (TACRO) + prednisone (PRED) +/- ipilimumab (IPI) for kidney transplant recipients (KTR) with advanced cutaneous cancers. Journal of Clinical Oncology 2022 40:16_suppl, 9507-9507) might incorporate the strategies employed in the current study in pursuit of that objective.

Response: Thank you for the valuable feedback. We added a brief summary of clinical trials investigating the safety and efficacy of ICI in kidney transplant recipients in the Discussion. The potential utility of our approach in clinical settings is also discussed (included below and in response to Reviewer 3's Point 4).

“Several prospective clinical studies are investigating strategies to mitigate acute rejection in kidney transplant recipients treated with ICI: continuing tacrolimus (NCT03816332), conversion to mammalian target of rapamycin (NCT04339062), or maintaining the baseline immunosuppression regimen^{24,25}. It is anticipated that results from these trials will help understand the underlining mechanisms of ICI-associated acute allograft rejection.”

“As the clonal frequency of alloreactive T cells is below detection at the pre-ICI timepoint in our experiments, it remains unclear whether an MLR preceding ICI therapy can expand and identify alloreactive T cells to predict future acute rejection upon exposure to ICI.”

Reviewer #2 (Remarks to the Author):

In this manuscript, Deepak A. Rao and co-workers have identified and tracked alloreactive T cells in a patient with melanoma who suffered from kidney transplant rejection following ICI therapy with the TCR sequencing. The results indicated that ICI therapy was related with an increase in circulating alloreactive CD8+ T cell clones. However, only TCR sequencing was performed. The conclusion was not reliable. More details should be added to investigate the mechanism about the increase of the CD8+ T cell clones.

Response: We thank the review for their assessment. Using a combination of an *in vitro* mixed-lymphocyte reaction experiment, paired scRNA/TCR-seq, and bulk TCR-seq, we identified a population of alloreactive CD8+ T cells present in the kidney and blood after the initiation of pembrolizumab. In efforts to further understand these cells, we have now performed scRNA/TCR-seq of the non-naïve CD8+ T cells in both post-pembrolizumab blood samples. In Figure 4, we show that the alloreactive population is present and is characterized expression of ZNF683, CXCR3, and HLA-DRA. These cells are highly proliferative early after pembrolizumab exposure. We also show this phenotype is overrepresented in clones that emerged in the blood after therapy compared to those present before therapy. These new data provide a critical new mechanistic insight into the identity of the alloreactive T cells activated by ICI therapy.

“After identifying and tracking alloreactive clones across tissues and over time, we next aimed to define the phenotypes of these alloreactive cells in the blood after pembrolizumab therapy. We sorted non-naive CD8+ T cells from both post-pembrolizumab blood samples and performed scRNA-seq to generate paired RNA and TCR libraries. Following clustering of the 14,186 CD8+ T cells from both samples, a combination of differential-gene expression analysis and comparison to recently published blood CD8+ T cell reference datasets was used to identify subpopulations (Figure 4A,B, Supplemental Figure 9A,B)¹⁴⁻¹⁶. Expected populations of cytotoxic T lymphocytes (CTLs), effector memory (EM), central memory (CM), and mucosal-associated invariant T (MAIT) cells were present. We also identified a population of cells marked by the strong expression of ZNF683 (encoding Hobit), CXCR3, and HLA-DRA, which we termed ZNF683+ T cells (Figure 4B). This cluster showed an elevated activation signature, potentially suggesting a set of cells that has become activated with the onset of ICI therapy (Figure 4C). Clustering analysis further identified small populations of actively proliferating cells and cells with elevated mitochondrial gene expression (Figure 4B, Supplemental Figure 9B). Deeper analysis of the proliferating cluster revealed a majority of included cells belong to the ZNF683+ subpopulation (Supplemental Figure 9D). Analysis of the composition of these populations between timepoints was consistent with our previous flow cytometry analysis, with increased CTLs and proliferating cells, and decreased EM cells, at Pembro 1 compared to Pembro 2 (Figure 4D, Supplemental Figure 6B,C, 9E).

Leveraging the paired TCR information for 13,034 (91.8%) of these non-naive CD8+ T cells, we sought to detect the cluster localizations of the MLR-defined alloreactive clones. Clones matching the alloreactive C1 MLR cluster (13/19) could be identified across both timepoints. Strikingly, at the Pembro 1 timepoint, these clones mapped predominately to the proliferating and ZNF683+ clusters, while at the Pembro 2 timepoint, the matching clones belonged almost exclusively to the ZNF683+ cluster (Figure 4E,F). In contrast, clones matching the C2 MLR cluster (12/15) were associated predominantly with CTLs, but could be identified in all clusters, with proportions that did not change across timepoints (Figure 4G, H). These results indicate that the MLR-defined alloreactive T cell clones possess a specific transcriptomic in the circulation that distinguishes them from the majority of circulating T cells.

To extend this analysis beyond the MLR-detected clones, we evaluated the phenotype of T cell clones that had newly emerged in the blood after pembrolizumab treatment using the bulk TCR data. Remarkably, clones were newly detected in the post-pembrolizumab blood samples but not present in pre-pembrolizumab samples (‘Emerged’) were highly overrepresented in the ZNF683+ cluster compared to clones that were stably present before and after pembrolizumab treatment (Figure 4I,J). Together, these highlight a specific phenotype of identified alloreactive clones as they become activated and enter the circulation following ICI initiation.”

We have also amended our Discussion to consider the implications of these additional data.

“It is difficult to define the phenotype of the alloreactive T cells expanded in an MLR because features will change with in vitro culture and activation; therefore, we used parallel scRNA-seq/TCR-seq of CD8+ T cells from the same blood samples to define the ex vivo phenotypes of T cells from the same clones. The MLR-defined alloreactive clones displayed a remarkably

homogeneous phenotype, characterized by expression of CXCR3 and ZNF683, with a transcriptomic signature that clearly distinguished these cells from the majority of circulating CD8+ T cells, including circulating cytotoxic, effector memory, and central memory cells. Beyond the MLR-defined clones, a larger set of clones that were detected in blood only after pembrolizumab also preferentially mapped to the ZNF683+ cluster, suggesting that this transcriptomic signature may capture a circulating set of cells that is characteristic of ICI therapy. The expression of ZNF683 in this cluster raises the possibility that these cells may be derived from tissue resident cells or possess some features of tissue residency¹⁸⁻²⁰.”

Together, we believe that the combination of in vitro experimentation, TCR sequencing to define alloreactive clones, and new single-cell profiling allows for a conceptual advance in our understanding of alloreactive T cells in transplant rejection, and provides methods to detect and interrogate these populations.

Reviewer #3 (Remarks to the Author):

This manuscript describes the expansion and distribution of alloreactive t-cells in a formally stable kidney transplant patient who develops cellular rejection after treatment with the ICI Pembrolizumab for malignant melanoma. T-cell clusters with an increased proliferation signal were identified using single cell sequencing of MLR expanded recipient PBMC's. An allo specific CD8 T cell clone was identified within cluster 1. Via TCR sequencing alloreactive clones were found to expand in peripheral blood and in the rejected kidney but not in a melanoma positive skin or lymph node biopsy.

This paper provides an interesting approach to the investigation of alloreactive t-cell dynamics in rejection in general and in specific for rejection induced by ICI. The potential of MLR to identify allo specific clones that can be followed in periferal blood is shown. This method offers a number of opportunities to further our understanding of the dynamics of alloreactivity. While the study is limited by the fact that only a single case is described and the findings are within the range of what would have been expected, the paper does show and interesting application of single cell technology in understanding rejection and may lead to meaningful follow up.

Response: We appreciate the reviewer's summary and positive comments.

I do have a few questions.

I could not find a statement on patient consent and ethics committee approval. I may have missed this. Otherwise please include this.

Response: We have included this statement and relevant institutional IRB number in the “Patient Samples” section of the methods.

“The patient consented to participate in the collection of biomedical specimens. The biobank study has been approved by Dana Farber Cancer Institute Institutional Review Board (IRB 05-042).”

2. The timing of the recipient sample for the initial MLR experiment is not clear to me. When were the samples obtained?

Response: Recipient PBMCs from the “Post-pembro 2” timepoint were used for the MLR experiment. We have included note of this in the “Mixed lymphocyte reaction” section of the methods.

“ 1×10^6 recipient PBMC (Pembro C2 timepoint) were loaded with CFSE (Invitrogen) and co-cultured with 4×10^6 irradiated donor splenocytes for 5 days in 10% FCS-RPMI, supplemented with recombinant human IL-2 (20 units/ml, PeproTech) and anti-CD28 (1 $\mu\text{g/ml}$) at 37C and 5% CO₂.”

3. The authors do not mention the possible clonal expansion of t-cells directed against the tumour. Did the authors also identify the expansion of clones that were not explained by alloreactivity but possibly by increased anti-tumor immunity?

Response: The review raises an excellent point with their interest in potential anti-tumor T cells to compare to the alloreactive T cells identified in this paper. While we could identify a number of T cell clones that expanded after the initiation of pembrolizumab and were not associated with the clones detected in our MLR experiment (Figure S5C), we cannot be certain that these T cells are reactive against the tumor. We do not see enrichment of these non-alloreactive clones in the tumor samples compared to kidney. The mixed-response of this patient to treatment may suggest a limited expansion of tumor-reactive T cells, but this remains speculative.

4. Do the authors envision that a initial MLR would help to predict the risk of rejection in patients who receive ICI?

Response: It is tempting to envision that the completion of an MLR before the initiation of immune checkpoint inhibitors could help to predict the risk of transplant rejection. In this work, we could not identify matching clones between the MLR-defined alloreactive cluster and the bulk TCR sequencing of pre-pembro blood, which includes over 48,500 unique T cell clones. This suggests that while these alloreactive clones may be present in blood at this time, they would be present at very low levels that may be challenging to detect technically. We have added points in the Discussion related to this.

“As the clonal frequency of alloreactive T cells is below detection at the pre-ICI timepoint in our experiments, it remains unclear whether an MLR preceding ICI therapy can expand and identify alloreactive T cells to predict future acute rejection upon exposure to ICI.”

Reviewer #4 (Remarks to the Author):

This study involves investigating alloreactive memory T cells in a patient with advanced melanoma who experienced allograft rejection after immune checkpoint inhibitor (ICI) therapy. The authors used single cell RNA and TCR sequencing, plus bulk RNA sequencing of TCR, to identify and track alloreactive T cells clones throughout the patients PD-1 treatment for melanoma. The authors conclude that they were able to identify a post ICI therapy subset of clonally expanded alloreactive CD8+ T cells.

What is noteworthy and unique about this particular study is that the tissues comes from a human patient with a clinically interesting case: stable conventional kidney transplant that develops advanced melanoma, is treated with ICI therapy and results in rejection of the transplant. Tissue samples were taken at key time points throughout the patients PD-1 therapy to longitudinally measure T cell clonality.

Response: We appreciate the reviewer’s comments on the novelty of this present work.

In Figure 2A the authors show a UMAP of single T cells collected from the MLR reaction – it shows 9 clusters that are grouped by CD8+ and CD4+ markers based on data they show in Supplemental data 1B-D. They then do further analysis using cell cycle markers to identify the proliferating CD8+ (cluster 1) population which they predict to be the alloreactive T cells. It is mentioned in the methods that differential expression with (Seurat-based FindMarkers function) and some of these markers are shown in Supplemental Figure 1B. I think the authors should show more of the DE genes (at least the top 5-10) as a heatmap and assign or discuss the biological significance of the remaining clusters. Especially Cluster 2 – it appears to be quite different from the other T cell populations. The cluster 2 population is referenced throughout the rest of the study (Figure 3, Supplemental Figure 7) so it would be helpful to try to describe the biological significance.

Response: Thank you for this suggestion to include more DEGs for each cluster in our MLR single-cell data. We have now included a supplementary heatmap that shows the top 10 markers for each cluster (S2C).

We have also added further discussion of the identities of several clusters in the Results.

“Among non-cycling clusters, C2 showed features of cytotoxicity (GZMB, PRF1, GNLY, GZMH), C7 had regulatory T cell markers (FOXP3, IKZF2), and C8 showed a high interferon signature (MX1, ISG16, OAS1) (Supplemental Figure 2C,D).”

We posited that the clusters identified following MLR and culture time may be altered by the in vitro culture and therefore not fully reflect the phenotype of these cells in the blood, so we generated an additional scRNA-seq/scTCR-seq dataset of non-naïve CD8+ T cells from both of the post-pembrolizumab blood samples. Here, we were able to connect MLR-defined clones to

their phenotype in the blood. With this, we noted that the alloreactive (C1) clones nearly exclusively belonged to a cluster with an activated phenotype characterized by expression of *ZNF683*, *CXCR3*, and *HLA-DRA* expression. In contrast, the C2 clones are predominately cytotoxic, and comprise the largest clones regardless of timepoint (Figures 4E-H).

“Leveraging the paired TCR information for 13,034 (91.8%) of these non-naive CD8+ T cells, we sought to detect the cluster localizations of the MLR-defined alloreactive clones. Clones matching the alloreactive C1 MLR cluster (13/19) could be identified across both timepoints. Strikingly, at the Pembro 1 timepoint, these clones mapped predominately to the proliferating and ZNF683+ clusters, while at the Pembro 2 timepoint, the matching clones belonged almost exclusively to the ZNF683+ cluster (Figure 4E,F). In contrast, clones matching the C2 MLR cluster (12/15) were associated predominantly with CTLs, but could be identified in all clusters, with proportions that did not change across timepoints (Figure 4G, H). These results indicate that the MLR-defined alloreactive T cell clones possess a specific transcriptomic in the circulation that distinguishes them from the majority of circulating T cells.”

The colors of the clusters are difficult to distinguish in the UMAPs in Figure 2A and Supplemental Figure 2B. Otherwise, I would label the clusters directly.

Response: We thank the reviewer for this suggestion. We have taken multiple steps to help the visualization of our clusters, including changing the cluster color palette to a set of colors more easily distinguishable and adding labels to the UMAP in Figure 2A.

For the UMAPs in Figure D2 showing expanded clonotypes, it would be helpful if cluster 1 and cluster 2 (major CD8+ populations) be outlined – similar to Figure 3C.

Response: We also amended the UMAPs in Figure 2D with outlines of clusters C1 and C2 (colored correspondingly), to aid in discerning the spatial localization of these clones.

Basic statistics for single cell RNA and TCR sequencing should be included in the methods or supplemental data: How many total cells are in this one sample? What is the percent of cells assigned clonotypes to single cells? What percentage of cells have an assigned clonotype for each cluster (resolution 0.7)?

Response: For the MLR data, we have added a supplementary figure (S2B) to highlight the total numbers of cells and those that pass QC.

In addition, we amended the text to help communicate the numbers of cells included in these analyses, and those with TCRs.

“Following dimensionality reduction of the 1,505 sorted cells that passed quality control filtering, we identified 9 clusters, including populations of CD8+ and CD4+ T cells (Figure 2A and Supplemental Figure 2B-F).”

“We then interrogated the paired TCR information available for 1,311 (87.1%) of these T cells. Both CD8+ T cell clusters (C1 and C2) contained expanded clones, while few expanded clones were found among CD4+ T cell populations (Figure 2C and Supplemental Figure 4A).”

Based on the reviewer’s suggestion, we have also included reporting of similar numbers for the additional single-cell sequencing dataset that was added for this revision (for example, Figure S9A).

The code for the single cell RNA analysis and TCR analysis, and the bulk TCR analysis should be submitted as supplementary information.

Response: We thank the reviewer for their attention to data access and reproducibility. The code used in the analysis of this data is now available on GitHub, and we have added a “Code availability” section of the methods.

“Scripts used for the analyses and figure generation of this paper are available at <https://github.com/dunlapg/transplant-rejection-ICI>.”

Database accession number for single cell RNA and TCR sequencing and bulk TCR sequencing was not provided with the manuscript.

Response: We have uploaded our sequencing data to GEO and have made note of the accession number in a “Data availability” section of the methods.

“Processed single-cell RNA/TCR sequencing output files from Cell Ranger and bulk TCR sequencing data generated by Adaptive Biotechnologies are available from the NCBI Gene Expression Omnibus (GEO) under accession GSE216763. Bulk TCR sequencing data can also be accessed using the Adaptive Biotechnologies ImmuneACCESS portal at <https://doi.org/10.21417/2022NC>.”

REVIEWER COMMENTS

Reviewer #1 (Remarks to the Author):

My comments have been addressed in the revised manuscript.

Reviewer #2 (Remarks to the Author):

1. In Figure 4B, the authors identified cells with elevated mitochondrial gene expression. For the single-cell RNA-seq data, high expression levels of mitochondrial genes could be an indicator of: 'Poor sample quality, leading to a high fraction of apoptotic or lysing cells'

2. As indicated in Supplemental Figure 2C, the C4, C5, C6, C7, C8, C9 clusters were not well identified. Especially the C4 and C5 clusters have almost same DEGs.

3. Trajectory analysis of the single-cell RNAseq profiles could help to illustrate the dynamics of alloreactivity.

Reviewer #3 (Remarks to the Author):

I am satisfied with the authors thoughtful response to my questions and remarks.

Reviewer #4 (Remarks to the Author):

The reviewers have sufficiently addressed my questions and suggestions.

REVIEWER COMMENTS

REVIEWER #2 (Remarks to the Author):

1. In Figure 4B, the authors identified cells with elevated mitochondrial gene expression. For the single-cell RNA-seq data, high expression levels of mitochondrial genes could be an indicator of: 'Poor sample quality, leading to a high fraction of apoptotic or lysing cells'

We agree that cells with signatures of high mitochondrial gene representation are associated with the death or lysing of those cells, and large numbers of such cells could be associated with diminished sample quality. As such, even though can find a cluster of cells with elevated mitochondrial content, these cells represent 0.5% of the overall samples both timepoints, suggesting samples of high quality. The overwhelming majority of cells in all other clusters, which are the focus of the analysis, have mitochondrial representation <10% and are dominated by cells of high quality.

Cluster	Pembro 1 Cells	Pembro 2 Cells
CTL	4588	4567
EM	701	1077
ZNF683+	216	272
CM	474	1112
MAIT	360	701
Cycling	32	15
Mito-hi	32	39
Total	6403	7783

2. As indicated in Supplemental Figure 2C, the C4, C5, C6, C7, C8, C9 clusters were not well identified. Especially the C4 and C5 clusters have almost same DEGs.

We have now incorporated lists of all differentially expressed genes for each cluster as supplementary tables for each dataset. We had previously added additional descriptions of the

key clusters in the first revision. While we could describe in detail each of these clusters, we think this adds little value to the manuscript and is likely to provide to readers a tedious description of clusters that are not relevant to the subsequent analyses. We hope that the full differential expression lists now included will provide sufficiently transparent material for interested readers to interrogate.

“After dimensionality reduction of the 1,505 sorted cells that passed quality control filtering, we identified 9 clusters, including populations of CD8⁺ and CD4⁺ T cells (Figure 2A, Supplemental Figure 2B-F, Supplemental Table 1).”

“Following clustering of the 14,186 CD8⁺ T cells from both samples, a combination of differential-gene expression analysis and comparison to recently published blood CD8⁺ T cell reference datasets was used to identify subpopulations (Figure 4A,B, Supplemental Figure 9A,B, Supplemental Table 4)¹⁴⁻¹⁶.”

“Among non-cycling clusters, C2 showed features of cytotoxicity (GZMB, PRF1, GNLY, GZMH), C7 had regulatory T cell markers (FOXP3, IKZF2), and C8 showed a high interferon signature (MX1, ISG16, OAS1) (Supplemental Figure 2C,D).”

3. Trajectory analysis of the single-cell RNAseq profiles could help to illustrate the dynamics of alloreactivity.

As suggested by the reviewer, we performed trajectory analysis of the scRNA-seq data of non-naïve CD8+ T cells. The resulting pseudotime trajectory largely follows a predictable course, indicating a progression of cells towards a terminal effector cytotoxic population, with the cluster enriched for alloreactive cells (light green below) in the middle. This is consistent with the overall orientation of the UMAP. Because the analysis provides little new insight beyond what would be expected from the UMAP itself, we prefer not to include this in the manuscript.

REVIEWERS' COMMENTS

Reviewer #2 (Remarks to the Author):

I have no further comments.

REVIEWER COMMENTS

REVIEWER #2 (Remarks to the Author):

I have no further comments.

We appreciate the reviewer's suggestions throughout the review process.